# Disparities in smokeless tobacco use in Bangladesh, India, and Pakistan: Findings from the Global Adult Tobacco Survey, 2014-2017

Luhua Zhao[1], Lazarous Mbulo[2]*, Evelyn Twentyman[2], Krishna Palipudi[2], Brian A. King[2]

**1** CDC Foundation, Atlanta, GA, United States of America, **2** Office on Smoking and Health, Centers for Disease Control and Prevention, Atlanta, GA, United States of America

\* vyp7@cdc.gov

**Data Availability Statement:** All data files are available from the GTSSdata base at https://www.cdc.gov/tobacco/global/gtss/gtssdata/index.html.

## Abstract

### Background

Smokeless tobacco (SLT) use is associated with multiple adverse health effects. It is prominent in Bangladesh, India, and Pakistan, but disparities in use within and across these countries are not well documented or understood. This study assessed the prevalence, patterns, and correlates of SLT use in these three countries.

### Method

Data came from the Global Adult Tobacco Survey, a household survey of adults aged ≥15 years. Data were collected in 2014 (Pakistan), 2017 (Bangladesh), and India (2016–2017). Current SLT use (nasal or oral use) was defined as reported SLT use daily or less than daily at the time of the survey. Prevalence of both overall and specific SLT types were assessed. Multivariate logistic regression was used to assess correlates of SLT use.

### Results

Overall, SLT use among adults ≥15 years of age was 20.6% in Bangladesh, 21.4% in India, and 7.7% in Pakistan, corresponding to 22.0 million SLT users in Bangladesh, 199.4 million in India, and 9.6 million in Pakistan. Among current tobacco users overall, the percentage of those who used SLT was 58.4% (CI: 56.0–60.7) in Bangladesh, 74.7% (CI: 73.4–76.0) in India, and 40.3% (CI: 36.2–44.5) in Pakistan. The most commonly used oral SLT product was Zarda (14.5%) in Bangladesh, Khaini (11.2%) in India, and Naswar (5.1%) in Pakistan. Females had greater odds of SLT use than males in Bangladesh, but lower odds of SLT use than males in India and Pakistan. In all three countries, the odds of SLT use was higher among those 25 years and older, lower education, lower wealth index, and greater exposure to SLT marketing.

**Funding:** Partial funding for the Global Adult Tobacco Survey (GATS) in Bangladesh, India, and Pakistan was provided by the Bloomberg Initiative to Reduce Tobacco Use through the CDC Foundation with a grant from Bloomberg Philanthropies.

**Competing interests:** The authors have declared that no competing interests exist.

## Conclusion

An estimated 231 million adults aged 15 years or older currently use SLT in Bangladesh, India, and Pakistan, comprising 40.3%-74.7% of overall tobacco product use in these countries. Moreover, marked variations in SLT use exist by population groups. Furthermore, exposure to pro-SLT marketing was found to be associated with higher SLT use compared to non-exposed. It is important that tobacco control strategies address all forms of tobacco product use, including SLT.

## Introduction

Smokeless tobacco (SLT) refers to non-combustible tobacco products, excluding e-cigarettes and heated tobacco products. SLT contains nicotine, which is highly addictive. Certain SLT products have also been found to contain carcinogens [1, 2], and evidence shows that SLT use can cause oral, pharyngeal, and esophageal cancers, and other dental diseases [3, 4]. SLT use has also been associated with increased risk for cardiovascular death and stillbirth [1, 5].

Sinha et al. [6] estimated that globally, the number of deaths that could be attributed to SLT, due to all causes, was 652,494 (234,008–1,081,437). This estimate excludes Europe, where studies showed no statistically significant association between all-cause mortality and SLT use. South East Asian Region carried the major proportion (88%) of this burden [7]."

SLTs are mainly available as oral and nasal products. Oral SLT products are either chewed, sucked in the mouth, placed between the gum and the cheeks, or applied to the gums or teeth directly. Nasal SLT products are applied through the nose. SLT is often mixed with other materials to enhance flavor for the user [4, 8]. More recently, novel SLT products have entered the market, such as dissolvable tablets or sticks made from finely milled tobacco, and toothpicks with tobacco-coating [1].

Although less well characterized in the public health literature than tobacco smoking, SLT use is common in many countries. To date, there are approximately 356.4 million people that use SLT across 140 countries [9]. Based on existing data, most SLT users live in low- and middle-income countries, with 237.3 million in India (2009–2010), 30.5 million in Bangladesh (2009), and 9.7 million in Pakistan (2014) for an approximate total of 277.5 million people using SLT [9].

In these countries, SLT use is often associated with culture and beliefs. For example, SLT use is often perceived to have health benefits, is widely socially accepted, and SLT products are offered at social events where smoking may not be considered socially acceptable [2, 10, 11].

Previous studies have assessed the prevalence and patterns of SLT use in south Asian countries. Palipudi et. al. [12] reported that among adults aged 15 years or older in Bangladesh in 2009, current SLT use was 27.2%. Data from Global Adult Tobacco Survey (GATS) India 2009–2010 and GATS Pakistan 2014 also showed that among persons aged 15 years or older, current SLT use was 25.9% and 7.7%, respectively [13]. However, understanding disparities in SLT use could help inform tobacco control strategies to reduce SLT-related morbidity and mortality. Previously released results have pointed to potential sex disparities within these countries, for example, current SLT use prevalence was 16.2% for males and 24.8% for females in Bangladesh during 2017 [14], 23.4% for males, and 12.3% for females in India during 2016–2017 [15], and 11.4% for males, and 3.7% for females in Pakistan during 2014 [15]. In addition,

it appears disparities may be present between urban and rural areas in India and Bangladesh but not in Pakistan [14–16].

However, there are limited studies that have examined disparities in the use of specific SLT products, including disparities by demographics and socioeconomic status. In addition, there are few studies on SLT use disparities associated with exposure to SLT-specific marketing, SLT-specific warnings, or self-reported awareness of SLT harms, which could influence use of these products. Therefore, this study analyzed the most recent GATS data from Bangladesh, India, and Pakistan to assess the prevalence, patterns, and disparities of SLT use, including concurrent and exclusive use of various types of SLT products, as well as correlates of SLT use, in each of these three countries.

## Methods

### Data source

This study utilized GATS data from Bangladesh, India, and Pakistan. In all three countries, GATS was implemented as a nationally representative, household-based, cross-sectional survey of non-institutionalized adults aged 15 years or older, which uses a standardized sample design, survey protocols, and questionnaire to ensure data comparability [17]. GATS was conducted in Bangladesh in 2017 with 14,880 completed individual interviews for a 90.8 overall response rate; India in 2016–2017 with 74,037completed individual interviews for a 92.9% overall response rate, and Pakistan in 2014 with 7,831 completed individual interviews for an overall response rate of 81.0%.

### Measures

**Tobacco product use.**   Current SLT use was defined with the question, "Do you currently use smokeless tobacco on a daily basis, less than daily, or not at all. Smokeless tobacco include zarda, sada pata, gul, khoinee, nosshil (Bangladesh); tobacco leaf, betel quid with tobacco, sada/surti, khaini or tobacco lime mixture, gutkha, pan masala with zarda, mawa, gul, gudaku, mishri (India); and naswar, nass (sniffed in the nose), paan with tobacco, gutka, mainpuri and others (Pakistan)". Using this question, we develop a new "current SLT use" variable consisting of individuals using SLT daily and less than daily.

Current tobacco smoking was defined with the question, "Do you currently smoke tobacco on a daily basis, less than daily, or not at all?" Using the question, we develop a new "current tobacco smoking" variable consisting of individuals who smoke tobacco on a daily and less than daily basis.

Current tobacco use was defined using the definition advanced by Kar, Sivanantham, Chinnakali, and Thiagarajan [18]: "Do you currently smoke tobacco?" and "Do you currently use smokeless tobacco?" Those who responded as "daily" and "less than daily" to both or either one of the questions, were defined as "current tobacco user" and those who responded, "not at all" were defined as "current tobacco nonuser" [18].

The use of specific SLT products was assessed using the following questions: 'On average, how many times a day (for daily users)/a week (for less than daily users) do you use the following product: [insert product name]?' SLT product types included in the questions are listed for all three countries in Table 2.

Current use of oral SLT was defined as a positive response to any of the SLT products other than 'Nasal use of snuff' in India and 'Naas' in Pakistan. Bangladesh did not collect information on nasal forms of SLT use.

**Demographic characteristics and socioeconomic status.**   Assessed demographic characteristics included: sex, urbanicity, age, and educational attainment. A wealth index was

constructed as a proxy measure of socio-economic status using information on household ownership of certain common household items such as electricity, flush toilet, fixed telephone, cell telephone, television, radio, refrigerator, car, moped/scooter/motorcycle, and washing machine [19]. The wealth index was divided into wealth index quintile rankings as follows: lowest, low, middle, high, and highest [19].

**Awareness of SLT harms.** A respondent was considered aware of the harms of SLT if they answered "yes" to the question, "Based on what you know or believe, does using smokeless tobacco cause serious illness?".

**Exposure to anti-SLT messages.** A respondent was considered to be exposed to an anti-SLT message if they answered "yes" to any of the following questions: "in the past 30 days, have you noticed any information in [insert media type] about the dangers of use or that encourages quitting of smokeless tobacco products". The types of media that were assessed across countries were: newspapers, magazines, television, radio, and billboards/hoardings; additionally, the Bangladesh survey included posters, and India included cinemas, the internet, public transportation vehicles, railways or bus stations, and public walls.

**Exposure to SLT marketing.** A respondent was considered to be exposed to SLT marketing if they answered "yes" to any of the questions that measured whether in the past 30 days the respondents noticed SLT advertisements and promotions in different venues. The assessed avenues included: stores where the products are sold, television, radio, billboards, posters, newspapers, magazines, cinemas, internet, public transportation vehicles or stations, and public walls.

## Analysis

Descriptive statistics, including point estimates and 95% confidence intervals (CI), were calculated for current SLT use, both overall and by SLT product type. Prevalence of current SLT use was calculated by selected sociodemographic characteristics. Differences between two estimates that are not independent of each other (e.g. prevalence of tobacco smoking vs. prevalence of SLT use given that a person can use both products) were tested using methods introduced by Wild and Seber [20], a variation of a normal test that takes into account overlapping cases and produces more precise variance. Estimates by sex and urbanicity were compared using Chi-square tests. Cochran–Mantel–Haenszel tests for trend were used to test for trends across age, education, and wealth index. T-tests were used for all other comparisons. For all analyses, p-values less than 0.05 were considered statistically significant.

Multivariate logistic regression was conducted to examine correlates of current SLT use; adjusted odds ratios (aOR) and corresponding 95% CI were calculated. Assessed correlates included: sex, urbanicity, age group, education, wealth index, current tobacco smoking status, awareness of SLT harms, exposure to anti-SLT messages, and exposure to SLT marketing.

SAS (Ver. 9.4) was used for data management and SAS-Callable SUDAAN (Ver. 11.0) was used for analyses in order to control for the complex survey design of GATS. Following GATS weighting protocol, all data were weighted to the estimated national target population in each respective country [21].

## Results

### Smokeless tobacco use–overall and by sex

Overall, prevalence of current SLT use was 20.6% (CI: 19.4–21.9) in Bangladesh, 21.4% (CI: 20.7–22.1) in India, and 7.7% (CI: 6.6–8.8) in Pakistan (Table 1). These prevalence estimates correspond to 22.0 (CI: 20.7–23.4) million current SLT users in Bangladesh, 199.4 (CI: 191.6–207.2) million in India, and 9.6 (CI: 8.2–10.9) million in Pakistan.

**Table 1. Sample size, prevalence and type of tobacco used overall and by sex among adults aged 15 years or older in Bangladesh, India and Pakistan, GATS 2014–2017.**

| | Bangladesh (2017) | | India (2016–17) | | Pakistan (2014) | |
|---|---|---|---|---|---|---|
| | Unweighted n | Weighted Percent (95% CI) | Unweighted n | Weighted Percent (95% CI) | Unweighted n | Weighted Percent (95% CI) |
| Total | | | | | | |
| Current Tobacco User | 12783 | 35.3 (33.9, 36.7) | 74037 | 28.6 (27.9, 29.3) | 7790 | 19.1 (17.4, 20.9) |
| Current Tobacco Smoker | 12783 | 18.0 (17.0, 19.0)* | 74037 | 10.7 (10.2, 11.1)* | 7831 | 12.4 (11.2, 13.8)* |
| Current Smokeless Tobacco User | 12783 | 20.6 (19.4, 21.9) | 74037 | 21.4 (20.7, 22.1) | 7780 | 7.7 (6.6, 8.8) |
| Nasal | - | - | 74037 | 0.5 (0.4, 0.6) | 7780 | 0.4 (0.2, 0.6) |
| Oral | 12783 | 20.6 (19.4, 21.9) | 74037 | 21.0 (20.3, 21.7) | 7781 | 7.3 (6.3, 8.5) |
| Current Smokeless Tobacco User among Current Tobacco Users | 5128 | 58.4 (56.0, 60.7) | 21857 | 74.7 (73.4, 76.0) | 1548 | 40.3 (36.2, 44.5) |
| Weighted number of SLT users (million) | 12783 | 22.0 (20.7, 23.4) | 74037 | 199.4 (191.6, 207.2) | 7780 | 9.6 (8.2, 10.9) |
| Male | | | | | | |
| Current Tobacco User | 6079 | 46.0 (43.9, 48.0) | 33772 | 42.4 (41.3, 43.5) | 3769 | 31.8 (28.8, 34.9) |
| Current Tobacco Smoker | 6079 | 36.2 (34.2, 38.2)** | 33772 | 19.0 (18.2, 19.9)** | 3782 | 22.2 (19.8, 24.8)** |
| Current Smokeless Tobacco User | 6079 | 16.2 (14.8, 17.7) | 33772 | 29.6 (28.7, 30.6) | 3759 | 11.4 (9.7, 13.4) |
| Nasal | - | - | 33772 | 0.6 (0.4, 0.7) | 3759 | 0.2 (0.1, 0.4) |
| Oral | 6079 | 16.2 (14.8, 17.7) | 33772 | 29.4 (28.4, 30.4) | 3760 | 11.2 (9.5, 13.2) |
| Current Smokeless Tobacco User among Current Tobacco Users | 3155 | 35.2 (32.5, 38.1) | 15576 | 69.9 (68.5, 71.3) | 1310 | 36.1 (31.8, 40.6) |
| Weighted Smokeless Tobacco users (million) | 6079 | 8.4 (7.7, 9.2) | 33772 | 141.2 (135.5, 146.8) | 3759 | 7.3 (6.1, 8.4) |
| Female | | | | | | |
| Current Tobacco User | 6704 | 25.2 (23.4, 27.1) | 40265 | 14.2 (13.5, 15.0) | 4021 | 5.8 (4.7, 7.0) |
| Current Tobacco Smoker | 6704 | 0.8 (0.5, 1.2) | 40265 | 2.0 (1.7, 2.3) | 4049 | 2.1 (1.6, 2.9) |
| Current Smokeless Tobacco User | 6704 | 24.8 (23.0, 26.6) | 40265 | 12.8 (12.0, 13.5) | 4021 | 3.7 (2.9, 4.8) |
| Nasal | - | - | 40265 | 0.5 (0.4, 0.6) | 4021 | 0.6 (0.3, 1.0) |
| Oral | 6704 | 24.8 (23.0, 26.6) | 40265 | 12.3 (11.5, 13.0) | 4021 | 3.2 (2.4, 4.3) |
| Current Smokeless Tobacco User among Current Tobacco Users | 1973 | 98.3 (97.1, 99.0) | 6281 | 89.8 (88.1, 91.2) | 238 | 64.4 (55.0, 72.7) |
| Weighted Smokeless Tobacco users (million) | 6704 | 13.6 (12.6, 14.6) | 40265 | 58.2 (54.5, 61.9) | 4021 | 2.3 (1.7, 2.8) |

Current tobacco smokers refer to those who reported smoking tobacco products daily or less than daily; Current smokeless Tobacco users refer to those who reported using smokeless tobacco products daily or less than daily. Current tobacco users refer to those were either a current tobacco smoker or current smokeless tobacco user.

-: Not available; N: Sample size; CI: Confidence interval.

* Wild & Seber tests $p < 0.05$ compared to that of SLT use from the same country.

** t-tests $p < 0.05$ compared to that of females from the same country.

SLT use was statistically significantly higher ($p < 0.05$) among females compared to males in Bangladesh (females, 24.8% CI: 23.0–26.6; males, 16.2% CI: 14.8–17.7), while current SLT use was significantly lower ($p < 0.05$) in females than males in both India (12.8% CI: 12.0–13.5 vs. 29.6% CI: 28.7–30.6) and Pakistan (3.7% CI: 2.9–4.8 vs. 11.4% CI: 9.7–13.4).

The prevalence of current SLT use was significantly higher ($p < 0.05$) than tobacco smoking in Bangladesh (20.6%; CI: 19.4–21.9 vs. 18.0%; CI: 17.0–19.0, respectively) and India (21.4%; CI: 20.7–22.1 vs. 10.7 CI: 10.2–11.1, respectively); in contrast, it was lower in Pakistan (7.7%; CI: 6.6–8.8 vs. 12.4%; CI: 11.2–13.8, respectively).

Among current tobacco users overall, the percentage of those who used SLT was 58.4% (CI: 56.0–60.7) in Bangladesh, 74.7% (CI: 73.4–76.0) in India, and 40.3% (CI: 36.2–44.5) in Pakistan (Table 1). In Bangladesh, 35.2% (CI: 32.5–38.1) of male current tobacco users were SLT users compared to 98.3% (CI: 97.1–99.0) of females. In India, the percentage was 69.9% (CI:

68.5–71.3) for males and 89.8% (CI:88.1–91.2) for females. In Pakistan, it was 36.1% (CI: 31.8–40.6) for males and 64.4% (CI: 55.0–72.7) for females.

## Smokeless tobacco use—by product type

The two most commonly used oral SLT types in Bangladesh were: betel quid with zarda, zarda only, or zarda with supari only (14.5%; CI: 13.5–15.6); and betel quid with sada pata only (5.5%; CI: 4.8–6.3). In India, the two most commonly used SLT types were: khaini only (11.2%; CI: 10.7–11.7); and gutka, areca nut-tobacco lime mixture, or mawa (6.8%; CI: 6.4–7.3). The two most commonly used SLT types in Pakistan were naswar only (5.1%; CI: 4.3–5.9); and paan with tobacco only (1.5%; CI: 1.1–2.0) (Fig 1).

Prevalence of nasal SLT use was 0.5% (CI: 0.4–0.6%) in India and 0.4% (CI: 0.2–0.6%) in Pakistan (Table 2). Bangladesh did not collect information on nasal SLT use.

## Smokeless tobacco use—by selected demographic and socioeconomic characteristics

**Urban/rural.** In Bangladesh, current SLT use was 22.5% (CI: 21.0–24.1) in rural areas compared to 14.9% (CI: 13.2–16.8) in urban areas. In India, it was 24.6% (CI: 23.8–25.4) in rural areas compared to 15.2% (CI:14.0–16.5) in urban areas. In Pakistan, it was 8.2% (CI: 6.7–10.1) in rural areas compared to 6.7% (CI: 5.6–7.9) in urban areas (Table 3). Prevalence of current SLT use was higher among rural residents than urban residents in Bangladesh and India (both p<0.05); no statistically significant difference was observed in Pakistan.

**Age.** Across all three countries, SLT use prevalence generally increased with increasing age (p<0.05). This disparity was particularly prominent in Bangladesh, where prevalence ranged from 4.0% (CI: 3.1–5.1) for those aged 15–24 years to 47.1% (CI: 42.1–52.3) for those aged 65+ years. In India, prevalence ranged from 10.8% (CI: 10.0–11.8) for those aged 15–24 years to 29.6% (CI: 27.6–31.7) for those aged 65+ years. In Pakistan, prevalence ranged from 3.4% (CI: 2.2–5.2) for those aged 15–24 years to 15.9% (CI: 11.9–21.0) for those aged 55–65 years.

**Education.** The prevalence of current SLT use decreased as education attainment increased in Bangladesh, India, and Pakistan (p<0.05). In Bangladesh, the prevalence ranged from 39.7% (CI:37.1–42.3) among those with no formal education to 5.9% (CI: 3.7–9.3) among those with college education or above (Table 3). In India, the prevalence ranged from 30.7% (CI: 29.0–32.5) among those with less than primary education to 7.5% (CI: 6.5–8.6) among those with college education or above. In Pakistan, the prevalence ranged from 10.6% (CI: 6.6–16.5) among those with less than primary school education, to 2.2% (CI: 1.1–4.5%) among those with college education or above.

**Wealth index.** The prevalence of current SLT use decreased as the wealth index increased in Bangladesh, India, and Pakistan (p<0.05) (Table 3). In Bangladesh, the prevalence ranged from 31.4% (CI: 29.0–34.0) among those in the lowest wealth index, to 10.8% (CI: 8.8–13.2) among those in the highest wealth index. In India, prevalence by wealth index ranged from 33.0% (CI: 31.6–34.4) among those in the lowest wealth index, to 7.3% (CI: 6.5–8.3) among those in the highest wealth index. In Pakistan, the respective prevalence ranged from 13.3% (CI: 9.7–17.9) among those in the lowest wealth index, to 3.0% (CI: 2.1–4.3) among those in the highest wealth index.

## Other tobacco product use

Current tobacco smokers had higher prevalence of SLT use compared to non-smokers (32.3%; CI: 30.3–34.3 vs. 20.1%; CI: 19.4–20.8, p<0.05) in India, but not in Bangladesh (18.3%; CI: 16.1–20.8 vs. 21.1%; CI: 19.8–22.5), and Pakistan (8.2%; CI: 5.7–11.6 vs. 7.6%; CI: 6.5–8.8).

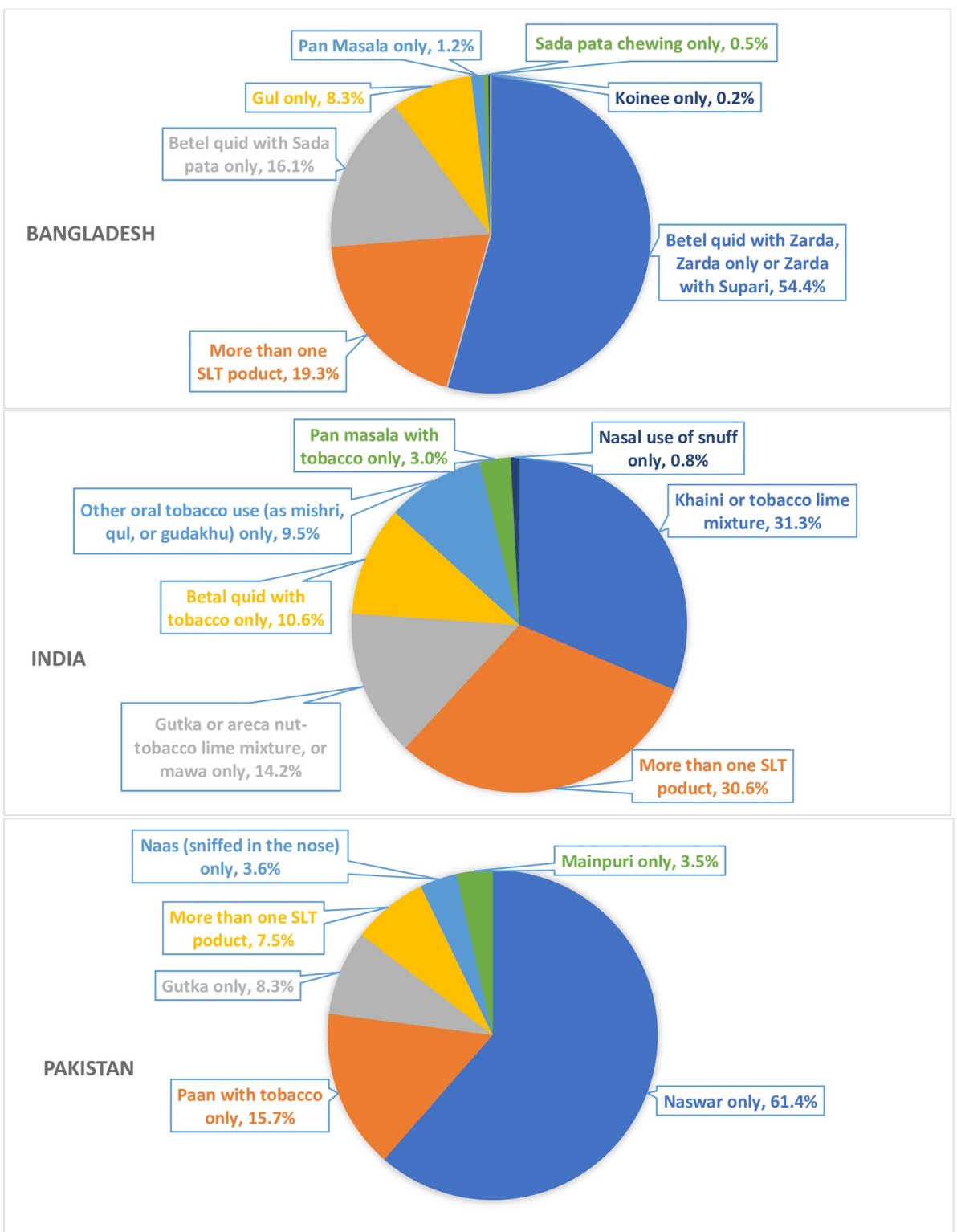

**Fig 1. Distribution of various smokeless tobacco products among current smokeless tobacco user in Bangladesh, India and Pakistan, GAT 2014–2017.**

**Table 2. Prevalence of current smokeless tobacco use overall and by type among adults aged 15+ by in Bangladesh, India and Pakistan, GATS 2014–2017.**

| | Bangladesh | | | India | | | Pakistan | |
|---|---|---|---|---|---|---|---|---|
| | Type | Percent (CI 95%) | | Type | Percent (CI 95%) | | Type | Percent (CI 95%) |
| Total | Overall SLT | 20.6 | (19.4, 21.9) | | 21.4 | (20.7, 22.1) | | 7.7 | (6.6, 8.8) |
| | Nasal | | | Nasal | | | Nasal | | |
| | N/A | - | - | Nasal use of snuff | 0.5 | (0.4, 0.6) | Naas (sniffed in the nose) | 0.4 | (0.2, 0.6) |
| | Oral | | | Oral | | | Oral | | |
| | Koinee | 0.1 | (0.0, 0.2) | Khaini or tobacco lime mixture | 11.2 | (10.7, 11.7) | Naswar | 5.1 | (4.3, 5.9) |
| | Pan Masala with tobacco | 0.8 | (0.6, 1.2) | Paan masala with tobacco | 2.8 | (2.6, 3.1) | Paan with tobacco | 1.5 | (1.1, 2.0) |
| | Gul | 3.6 | (3.1, 4.3) | Gutka, areca nut-tobacco lime mixture, or Mawa | 6.8 | (6.4, 7.3) | Gutka | 0.8 | (0.5, 1.4) |
| | Sada pata chewing | 0.7 | (0.4, 1.0) | Oral tobacco use (as mishri, qul, or gudakhu) | 3.8 | (3.5, 4.1) | Mainpuri | 0.4 | (0.1, 0.9) |
| | Betel quid with Sada pata | 5.5 | (4.8, 6.3) | Betal quid with tobacco | 5.8 | (5.4, 6.2) | | | |
| | Betel quid with Zarda, Zarda only, or Zarda with Supari | 14.5 | (13.5, 15.6) | | | | | | |
| | Other unspecified SLT | 0.1 | (0.0, 0.4) | Other unspecified SLT | 0.3 | (0.2, 0.3) | Other unspecified SLT | 0.2 | (0.1, 0.3) |
| Male | Overall SLT | 16.2 | (14.8, 17.7) | | 29.6 | (28.7, 30.6) | | 11.4 | (9.7, 13.4) |
| | Nasal | | | Nasal | | | Nasal | | |
| | N/A | - | - | Nasal use of snuff | 0.7 | (0.5, 0.9) | Naas (sniffed in the nose) | 0.2 | (0.1, 0.4) |
| | Oral | | | Oral | | | Oral | | |
| | Koinee | 0.1 | (0.0, 0.3) | Khaini or tobacco lime mixture | 17.9 | (17.0, 18.7) | Naswar | 8.4 | (7.0, 9.9) |
| | Pan Masala with tobacco | 0.3 | (0.2, 0.5) | Paan masala with tobacco | 4.5 | (4.0, 5.0) | Paan with tobacco | 1.7 | (1.1, 2.7) |
| | Gul | 3.1 | (2.4, 4.0) | Gutka, areca nut-tobacco lime mixture, or Mawa | 10.8 | (10.1, 11.5) | Gutka | 1.3 | (0.8, 2.0) |
| | Sada pata chewing | 0.3 | (0.1, 0.7) | Oral tobacco use (as mishri, qul, or gudakhu) | 3.3 | (3.0, 3.7) | Mainpuri | 0.5 | (0.2, 1.4) |
| | Betel quid with Sada pata | 2.3 | (1.8, 3.0) | Betal quid with tobacco | 7.1 | (6.5, 7.7) | | | |
| | Betel quid with Zarda, Zarda only, or Zarda with Supari | 13.0 | (11.7, 14.3) | | | | | | |
| | Other unspecified SLT | 0.0 | N/A | Other unspecified SLT | 0.3 | (0.2, 0.4) | Other unspecified SLT | 0.3 | (0.2, 0.6) |
| Female | Overall SLT | 24.8 | (23.0, 26.6) | | 12.8 | (12.0, 13.5) | | 3.7 | (2.9, 4.8) |
| | Nasal | | | Nasal | | | Nasal | | |
| | N/A | - | - | Nasal use of snuff | 0.6 | (0.5, 0.7) | Naas (sniffed in the nose) | 0.6 | (0.3, 1.0) |
| | Oral | | | Oral | | | Oral | | |
| | Koinee | 0.0 | (0.0, 0.1) | Khaini or tobacco lime mixture | 4.2 | (3.8, 4.6) | Naswar | 1.7 | (1.2, 2.3) |
| | Pan Masala with tobacco | 1.3 | (0.8, 2.1) | Paan masala with tobacco | 1.1 | (0.9, 1.3) | Paan with tobacco | 1.2 | (0.8, 1.9) |
| | Gul | 4.1 | (3.4, 5.0) | Gutka, areca nut-tobacco lime mixture, or Mawa | 2.7 | (2.3, 3.1) | Gutka | 0.3 | (0.1, 1.1) |
| | Sada pata chewing | 1.0 | (0.6, 1.7) | Oral tobacco use (as mishri, qul, or gudakhu) | 4.3 | (3.9, 4.8) | Mainpuri | 0.2 | (0.1, 0.4) |
| | Betel quid with Sada pata | 8.5 | (7.3, 9.8) | Betal quid with tobacco | 4.5 | (4.0, 5.0) | | | |
| | Betel quid with Zarda, Zarda only, or Zarda with Supari | 16.0 | (14.6, 17.6) | | | | | | |
| | Other unspecified SLT | 0.1 | (0.0, 0.7) | Other unspecified SLT | 0.3 | (0.2, 0.4) | Other unspecified SLT | 0.0 | (0.0, 0.1) |

N/A: Not available; -: Data not available. SLT: Smokeless tobacco. CI: Confidence Interval.

**Table 3. Prevalence and adjusted odds ratios of current smokeless tobacco use among adults aged 15 years or older by selected demographic and socioeconomic characteristics in Bangladesh, India and Pakistan, GATS 2014–2017.**

| | | Bangladesh | | | India | | | Pakistan | | |
|---|---|---|---|---|---|---|---|---|---|---|
| | | N | Percent (95% CI) | aOR (95% CI) | N | Percent (95% CI) | aOR (95% CI) | N | Percent (95% CI) | aOR (95% CI) |
| | Overall | 12783 | 20.6 (19.4, 21.9) | - | 74037 | 21.4 (20.7, 22.1) | - | 7780 | 7.7 (6.6, 8.8) | - |
| Sex | Male | 6079 | 16.2 (14.8, 17.7) | REF | 33772 | 29.6 (28.7, 30.6) | REF | 3759 | 11.4 (9.7, 13.4) | REF |
| | Female | 6704 | 24.8 (23.0, 26.6) # | 1.53 (1.31, 1.80) * | 40265 | 12.8 (12.0, 13.5) # | 0.26 (0.24, 0.28) * | 4021 | 3.7 (2.9, 4.8) # | 0.20 (0.14, 0.28) * |
| Urbanicity | Urban | 6356 | 14.9 (13.2, 16.8) | REF | 26488 | 15.2 (14.0, 16.5) $ | REF | 3770 | 6.7 (5.6, 7.9) | REF |
| | Rural | 6427 | 22.5 (21.0, 24.1) # | 1.15 (0.96, 1.37) | 47549 | 24.6 (23.8, 25.4) # | 1.18 (1.05, 1.32) * | 4010 | 8.2 (6.7, 10.1) | 0.84 (0.61, 1.15) |
| Age group | 15–24 | 2345 | 4.0 (3.1, 5.1) $ | REF | 13329 | 10.8 (10.0, 11.8) $ | REF | 2095 | 3.4 (2.2, 5.2) $ | REF |
| | 25–34 | 3363 | 14.5 (12.9, 16.3) | 3.53 (2.69, 4.64)* | 18600 | 21.2 (20.1, 22.3) | 2.29 (2.04, 2.56)* | 2015 | 8.5 (6.5, 10.9) | 3.13 (1.88, 5.21)* |
| | 35–44 | 3034 | 23.2 (20.8, 25.8) | 5.93 (4.46, 7.88)* | 16964 | 25.7 (24.4, 27.0) | 2.85 (2.51, 3.24)* | 1658 | 8.2 (6.5, 10.3) | 2.99 (1.72, 5.18)* |
| | 45–54 | 2088 | 36.4 (33.1, 39.9) | 10.35 (7.73, 13.86)* | 11501 | 27.0 (25.6, 28.6) | 3.03 (2.62, 3.50)* | 955 | 10.4 (8.5, 12.7) | 4.36 (2.50, 7.57)* |
| | 55–64 | 1124 | 41.5 (37.4, 45.7) | 12.52 (9.20, 17.04)* | 7631 | 27.3 (25.7, 28.9) | 2.78 (2.38, 3.24)* | 602 | 15.9 (11.9, 21.0) | 5.68 (2.92, 11.03)* |
| | 65+ | 829 | 47.1 (42.1, 52.3) | 15.10 (10.75, 21.21)* | 6012 | 29.6 (27.6, 31.7) | 3.08 (2.62, 3.62)* | 455 | 8.8 (5.8, 13.0) | 3.38 (1.65, 6.94)* |
| Education | No formal education | 3581 | 39.7 (37.1, 42.3) $ | REF | 18473 | 28.9 (27.7, 30.1) $ | REF | 3597 | 10.1 (8.4, 12.0) $ | REF |
| | Less than primary | 2057 | 24.2 (21.8, 26.7) | 0.78 (0.65, 0.93) * | 7510 | 30.7 (29.0, 32.5) | 0.92 (0.83, 1.02) | 393 | 10.6 (6.6, 16.5) | 1.10 (0.60, 2.03) |
| | Primary completed | 1573 | 16.2 (13.8, 19.0) | 0.59 (0.48, 0.73) * | 8858 | 26.8 (25.3, 28.4) | 0.84 (0.75, 0.93) * | 806 | 7.6 (5.3, 10.6) | 0.79 (0.53, 1.18) |
| | Less than secondary | 2710 | 10.7 (9.3, 12.3) | 0.45 (0.36, 0.56) * | 12109 | 22.3 (20.9, 23.7) | 0.75 (0.67, 0.83) * | 739 | 8.6 (5.7, 12.7) | 0.94 (0.56, 1.59) |
| | Secondary/high school completed | 2059 | 6.1 (4.7, 7.9) | 0.32 (0.23, 0.44) * | 18290 | 13.1 (12.2, 14.1) | 0.46 (0.41, 0.52) * | 1633 | 3.5 (2.6, 4.8) | 0.43 (0.29, 0.64) * |
| | College and above | 803 | 5.9 (3.7, 9.3) | 0.23 (0.14, 0.39) * | 8738 | 7.5 (6.5, 8.6) | 0.27 (0.23, 0.33) * | 611 | 2.2 (1.1, 4.5) | 0.26 (0.11, 0.61) * |
| Wealth index | Lowest | 2561 | 31.4 (29.0, 34.0) $ | REF | 15547 | 33.0 (31.6, 34.4) $ | REF | 1469 | 13.3 (9.7, 17.9) $ | REF |
| | Low | 2585 | 22.6 (20.5, 25.0) | 0.71 (0.59, 0.85) * | 18685 | 24.5 (23.4, 25.7) | 0.69 (0.63, 0.76) * | 1690 | 10.5 (8.3, 13.3) | 0.63 (0.42, 0.93) * |
| | Medium | 2525 | 20.7 (18.4, 23.2) | 0.75 (0.62, 0.91) * | 11278 | 20.5 (19.1, 22.0) | 0.59 (0.52, 0.66) * | 1559 | 6.6 (5.1, 8.5) | 0.56 (0.34, 0.91) * |
| | High | 2571 | 17.0 (14.8, 19.4) | 0.63 (0.50, 0.78) * | 14814 | 13.9 (12.9, 15.1) | 0.42 (0.37, 0.48) * | 1064 | 7.4 (5.2, 10.2) | 0.63 (0.36, 1.10) |
| | Highest | 2541 | 10.8 (8.8, 13.2) | 0.44 (0.33, 0.58) * | 13713 | 7.3 (6.5, 8.3) | 0.25 (0.21, 0.30) * | 1998 | 3.0 (2.1, 4.3) | 0.23 (0.12, 0.41) * |
| Current tobacco smoking status | Yes | 2493 | 18.3 (16.1, 20.8) | 0.65 (0.54, 0.80) * | 9499 | 32.3 (30.3, 34.3) | 0.73 (0.65, 0.82) * | 953 | 8.2 (5.7, 11.6) | 0.39 (0.24, 0.65) * |
| | No | 10290 | 21.1 (19.8, 22.5) | REF | 64538 | 20.1 (19.4, 20.8) # | REF | 6827 | 7.6 (6.5, 8.8) | REF |
| Awareness of SLT harm | Yes | 12178 | 20.4 (19.2, 21.8) | 1.05 (0.77, 1.43) | 70798 | 21.0 (20.3, 21.7) | 0.80 (0.69, 0.91)* | 5923 | 7.5 (6.5, 8.8) | 1.00 (0.74, 1.37) |

(*Continued*)

**Table 3.** (Continued)

| | | Bangladesh | | | India | | | Pakistan | | |
|---|---|---|---|---|---|---|---|---|---|---|
| | | N | Percent (95% CI) | aOR (95% CI) | N | Percent (95% CI) | aOR (95% CI) | N | Percent (95% CI) | aOR (95% CI) |
| | No | 602 | 23.6 (18.8, 29.1) | REF | 3221 | 29.5 (27.0, 32.2) [#] | REF | 1802 | 8.0 (6.4, 10.0) | REF |
| Exposure to anti-SLT messages | Yes | 4240 | 20.8 (18.6, 23.1) | 1.46 (1.24, 1.71) * | 48588 | 20.0 (19.2, 20.8) | 1.00 (0.92, 1.09) | 1242 | 5.4 (3.9, 7.5) | 0.61 (0.40, 0.95) * |
| | No | 8492 | 20.4 (19.1, 21.8) | REF | 25414 | 24.3 (23.2, 25.4) [#] | REF | 6168 | 8.1 (7.0, 9.5) | REF |
| Exposure to SLT marketing | Yes | 1006 | 28.4 (24.2, 33.0) | 1.78 (1.43, 2.23) * | 13507 | 25.3 (23.8, 27.0) | 1.49 (1.35, 1.65) * | 906 | 13.6 (10.6, 17.4) | 2.17 (1.57, 3.00) * |
| | No | 11770 | 20.0 (18.8, 21.2) [#] | REF | 60411 | 20.4 (19.7, 21.1) [#] | REF | 6172 | 6.8 (5.7, 8.0) [#] | REF |

REF: Reference group. aOR: Adjusted odds ratio. CI: Confidence Interval. aOR and CI for aOR retain two decimals for greater precision.

* p value for Wald F test <0.05 from logistic regression.

[#] Chi square tests p<0.05.

[$] Cochran-Mantel-Haenszel tests for trend p<0.05. SLT: Smokeless tobacco.

## Knowledge of SLT harms

In India, those who were aware of SLT harms had lower prevalence of SLT use compared to those who were not aware (21.0%; CI: 20.3–21.7 vs. 29.5%; CI: 27.0–32.2. p<0.05), but not in Bangladesh (20.4%; CI: 19.2–21.8 vs. 23.6%; CI: 18.8–29.1), and Pakistan (7.5%; CI: 6.5–8.8 vs. 8.0%; CI: 6.4–10.0).

## Pro and anti SLT messaging

In India, those who were exposed in the past 30 days to anti-SLT use messages had lower SLT prevalence compared to those who were not exposed (20.0%; CI: 19.2–20.8%) vs. 24.3%; CI: 23.2–25.4%) (P<0.05) but not in Bangladesh (20.8%; CI: 18.6–23.1) vs. 20.4%; CI: 19.1–21.8)) and Pakistan (5.4%; CI: 3.9–7.5%) vs. 8.1%; CI: 7.0–9.5%).

In all three countries, those who were exposed in the past 30 days to pro-SLT marketing had higher SLT use prevalence compared to those who were not exposed (p<0.05). The prevalence for Bangladesh was 28.4% (CI:24.2–33.0%) for those exposed compared to 20.0% (CI: 18.8–21.2%) for those not exposed. In India, prevalence was 25.3% (23.8–27.0) compared to 20.4% (CI: 19.7–21.1%). In Pakistan, prevalence was 13.6% (CI: 10.6–17.4%) compared to 6.8% (CI: 5.7–8.0%).

## Multivariate analyses

The odds of SLT use was higher among females than males in Bangladesh (aOR: 1.53; CI: 1.31–1.80), but lower in females than males in India (aOR: 0.26; CI: 0.24–0.28) and Pakistan (aOR: 0.20; CI: 0.14–0.28). Compared to those aged 15–24 years, the odds of SLT use in Bangladesh was higher among those aged 25–45 years (aOR: 4.30; CI: 3.34–5.62), 45–65 years (aOR: 9.80; CI: 7.40–12.86), and 65+ years (aOR: 12.80; CI 9.15–17.96). In India, compared to those aged 15–24 years, the odds of SLT use was higher among those aged 25–45 years (aOR: 2.40; CI: 2.19–2.73), 45–65 years (aOR: 2.70; CI: 2.33–3.04), and 65+ years (aOR: 2.8 (2.36–3.24). In Pakistan, compared to those aged 15–24 years, the odds of SLT use was higher among those aged 25–45 years (aOR: 2.40; CI: 1.47–3.76), 45–65 years (aOR: 3.70; CI: 2.23–6.20), and 65+ years (aOR: 2.20, CI: 1.15–4.19).

In Bangladesh, compared to those with no formal education, the odds of SLT use were lower among those with less than primary (aOR: 0.78; CI 0.65–0.93), primary completed (aOR: 0.59; CI: 0.48–0.73), less than secondary (aOR: 0.45; CI:0.36–0.56), secondary/high school complete (aOR:0.32; CI: 0.23–0.44), and college and above (aOR: 0.23; CI: 0.14–0.39). In India, compared to those with no formal education, the odds of SLT use were lower among those with primary completed (aOR: 0.84; CI: 0.75–0.93), less than secondary (aOR: 0.75; CI:0.67–0.83), secondary/high school complete (aOR:0.46; CI: 0.41–0.52), and those with college and above (aOR: 0.27; CI: 0.23–0.33). In Pakistan, compared to those with no formal education, the odds of SLT use were lower among those with secondary/high school complete (aOR:0.43; CI: 0.29–0.64), and those with college and above (aOR: 0.26; CI: 0.11–0.61).

Those ranked in the highest wealth index in all three countries had lower odds of SLT use compared to those in the lowest wealth index. In Bangladesh, compared to the lowest wealth index, the odds of SLT use were lower among those in the low (aOR:0.71; CI: 0.59–0.85), medium (aOR: 0.75; CI:0.62–0.91), high (aOR: 0.63: CI: 0.50–0.78), and highest (aOR: 0.44; CI: 0.33–0.58) wealth indices. In India, compared to the lowest wealth index, the odds of SLT use were lower among those in the low (aOR:0.69; CI: 0.63–0.76), medium (aOR: 0.59; CI: 0.52–0.66), high (aOR: 0.42: CI: 0.37–0.48), and highest (aOR: 0.25; CI: 0.21–0.30) wealth indices. In Pakistan, compared to the lowest wealth index, the odds of SLT use were lower among those in the low (aOR: 0.63; CI: 0.42–0.93), medium (aOR: 0.56; CI: 0.34–0.91), and highest (aOR: 0.3; CI: 0.15–0.55) wealth indices.

In all three countries, the odds of SLT use were lower among current tobacco smokers compared to non-tobacco smokers. Compared to non-tobacco smokers, the odds of SLT use among current tobacco smokers was lower in Bangladesh (aOR: 0.65; CI: 0.54–0.80), India (aOR: 0.73; CI: 0.65–0.82), and Pakistan (aOR: 0.39; (0.24–0.65).

In Bangladesh, the odds of SLT use were higher among those exposed to anti-SLT messages in the past 30 days compared to those not exposed (aOR: 1.46; CI: 1.24–1.71); in India, no significant association between exposure to anti-SLT messages in the past 30 days and SLT use were observed; and in Pakistan, the odds of SLT use were lower among those exposed to anti-SLT messages in the past 30 days compared to those not exposed (aOR: 0.61; CI: 0.40–0.95). Awareness of SLT harms was only significant (p<0.05) in India, where the odds of STL use among those aware of the harms of SLT use was higher compared to those not aware (aOR: 0.80; CI: 0.69–0.91).

## Discussion

The findings from this study reveal that current SLT use comprises a large portion of overall tobacco use in the assessed countries, including nearly 6 in 10 persons who currently use tobacco in Bangladesh, more than 7 in 10 in India, and about 4 in 10 in Pakistan. Across all three countries, current SLT use was higher among the 25 years and older age groups, particularly in Bangladesh and India. Additionally, all three countries showed a marked socio-demographic and economic disparity in SLT use defined by sex, education, wealth index, and age. Specifically, adults with lower education levels, and adults with lower wealth index had significantly higher odds of current SLT use. These findings are consistent with those in other studies, where use of SLT was high among those with low SES [12, 22, 23]. It is therefore important that these underlying socio-demographic, economic, and/or environmental disadvantages are considered when implementing SLT prevention and reduction strategies.

The challenge to addressing tobacco use among low SES population in Bangladesh, India, and Pakistan may also need to address the existence of tax evasion, illicit sales and production of smokeless tobacco including illicit trade and low levels of taxation for these products [11,

24]. This illegal supply chain provide affordable and accessible SLT products that sustains the consumption of these products among the low SES population. Addressing this problem could help in restricting availability and affordability of SLT products to low SES population and young people critical to preventing and reducing consumption.

In addition, the study found socio-demographic differences in current SLT use by sex that were apparent in each country. Females had lower odds of current SLT use in India and Pakistan compared to males, but higher odds in Bangladesh. This finding is consistent with a previous study in Bangladesh [25] and cross-country sex differences may be the result of variations in social acceptability of tobacco product use [2, 3, 26]. This suggest the need for SLT use prevention and cessation strategies to take into consideration historical, social, and cultural acceptance of SLT use in all three countries, even among females, that might also be driving some of the observed disparities [23]. Another consideration is that the odds of current SLT use was lower among current tobacco smokers compared to non-tobacco smokers. Thus, Bangladesh, India, and Pakistan could consider addressing SLT use separately in tobacco control efforts given that the economic and health effects of SLT use are different from that of smoking [24] in most low-resource and high SLT burden Parties has been reported in the MPOWER 2017, which is required to be strengthene.

Furthermore, our results showed that exposure to pro-SLT marketing was associated with higher SLT use compared to non-exposed. As parties to WHO Framework Convention on Tobacco Control (FCTC), all three countries have at various levels implemented bans on tobacco advertising, promotion and sponsorship (FCTC Article 13) particularly a ban on product display which, is main advertisement tool for point-of-sale vendors [27, 28]. India has passed comprehensive ban on advertising, promotion, and sponsorship of all tobacco products including SLT [28]. Bangladesh made amendment to Tobacco Control Law in 2013 to require graphic health warnings to cover 50% of SLT packaging, ban on advertisement of SLT products, and restriction to sale to minors [11]. Pakistan has also passed legislations on tobacco control that could indirectly affect SLT production, sale, promotion, and consumption. However, a lack of specific wordings in the legislations in Pakistan for SLT, raise challenges with enforcement of the law [28]. In all three countries, there are difficulties in enforcement of the law banning SLT promotion and sponsorship [11, 28].

Finally, this study confirms that the majority of SLT products consumed in all three countries were in oral form and shows that use of nasal tobacco is relatively low in both India and Pakistan [1]. Although some SLT products are common (e.g. gul or gutka) in Bangladesh, India, and Pakistan, the most commonly used products differed in these three countries [1]. This suggests the importance of addressing the significant heterogeneity of SLT products and their toxic constituents and additives, and evidence-based strategies for SLT use prevention and control. Such strategies could include SLT product surveillance and monitoring, establishing effective and relevant health warning labels on SLT products, and cessation support [1]. Given a positive association was found for exposure to anti-SLT messages and SLT use in Bangladesh, this result may suggest ongoing review and updating of public health messaging campaigns around SLT. Other interventions may include establishing standards for toxicants and maximum pH levels, effective health warning labels, increasing prices on SLT products, prohibiting SLT promotion, sponsorship, or marketing [1].

This study is subject to some limitations. First, data were self-reported, which could introduce recall biases or underestimates of SLT use due to social desirability biases. However, GATS use a standardized global protocol to produce nationally representative estimates with measures that are comparable across countries [21]. Second, although all three countries used the same protocol and questions, there are variations in data collection time across the

countries that might warrant caution in cross-country comparisons. Finally, we used the wealth index as a proxy for socio-economic status. However, the wealth index is considered an accepted proxy for socio-economic status in household surveys [29].

## Conclusion

SLT use comprises the most tobacco use in Bangladesh (58.4%), and India (74.7%), and a large portion of tobacco use in Pakistan (40.3%), with clear disparities in use by socio-demographic and economic characteristics in these countries. Importantly, SLT use remain a major public health challenge in the three countries and other South and Southeast Asian countries which, suggests a need to prioritize SLT in tobacco control efforts [23]. It may be beneficial to focus STL use prevention and control interventions at populations with high STL use prevalence such as older adults, those with lower education, and those in the lower wealth quintile. In addition, all three countries may need to focus on SLT use among females as majority of females who used tobacco were using SLT. Finally, our findings demonstrate that opportunities exist to improve anti-SLT messaging, reduce exposure to SLT marketing, and protecting populations with a higher prevalence of SLT use in these countries.

## Acknowledgments

**Disclaimer:** The findings and conclusions in this report are those of the authors and do not necessarily represent the official position of the U.S. Centers for Disease Control and Prevention.

## Author Contributions

**Conceptualization:** Luhua Zhao, Lazarous Mbulo, Evelyn Twentyman, Krishna Palipudi, Brian A. King.

**Formal analysis:** Luhua Zhao.

**Methodology:** Luhua Zhao, Lazarous Mbulo, Evelyn Twentyman, Krishna Palipudi, Brian A. King.

**Writing – original draft:** Lazarous Mbulo, Evelyn Twentyman, Krishna Palipudi, Brian A. King.

**Writing – review & editing:** Luhua Zhao, Lazarous Mbulo, Evelyn Twentyman, Krishna Palipudi, Brian A. King.

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
