## [Decision Letter · Decision Letter 0]

10 Feb 2021

PONE-D-21-01174

Disparities in Smokeless Tobacco Use in Bangladesh, India, and Pakistan: Findings from the Global Adult Tobacco Survey, 2014 -2017

PLOS ONE

Dear Dr. Mbulo,

Thank you for submitting your manuscript to PLOS ONE. After careful consideration, we feel that it has merit but does not fully meet PLOS ONE’s publication criteria as it currently stands. Therefore, we invite you to submit a revised version of the manuscript that addresses the points raised during the review process.

We look forward to receiving your revised manuscript.

Kind regards,

Pranil Man Singh Pradhan

Academic Editor

PLOS ONE

Additional Editor Comments:

Production, import and sale of illicit smokeless tobacco products are also contributory factors fueling the SLT epidemic in these countries. Although the objectives of this study does not directly relate to it, it would be worthwhile to mention this in the discussion.

Journal Requirements:

2. Please amend either the abstract on the online submission form (via Edit Submission) or the abstract in the manuscript so that they are identical.

Reviewers' comments:

Reviewer's Responses to Questions

**Comments to the Author**

1. Is the manuscript technically sound, and do the data support the conclusions?

Reviewer #1: Yes

Reviewer #2: Yes

2. Has the statistical analysis been performed appropriately and rigorously? 

Reviewer #1: No

Reviewer #2: Yes

3. Have the authors made all data underlying the findings in their manuscript fully available?

Reviewer #1: No

Reviewer #2: Yes

4. Is the manuscript presented in an intelligible fashion and written in standard English?

Reviewer #1: Yes

Reviewer #2: Yes

5. Review Comments to the Author

Reviewer #1: Dear authors,

It was an interesting read. however going through the article I found some issues.

Introduction:

• In line 71 and 73 the disease burden due to SLT use is explained in DALYs lost. It is confusing as one DALY means a year of healthy life lost. For eg: disease burden due to SLT use was 8,691,827 disability-71 adjusted life years (DALYs) and 72 348,798 deaths sounds appropriate to me.

• In line 98 author have said no literature have examined disparities in SLT use however in a paragraph just above in line 93 to 97 have presented the findings from GATS survey which have found different disparities in SLT use.

• The authors have said that the study analyzed the data from GATS survey and assessed the prevalence

Methodology:

• The authors have stated using the data from GATS survey at different time period in three south Asian countries. In line 111 the sample size and response rate of the study was according to the GATS survey itself, or authors identified is confusing to understand. The data collection method is not clear, how the data were extracted for analysis and presentation needs a clearer explanation.

• Different measures are explained as means of data collection but were these measures an operational definition from GATS survey or authors of the study created it? If it is from GATS survey than it is to be explained as the measures taken in survey to generate data in different variables were: - OR if it came from author of this article it should explain on more how data are presented in GATS survey and how authors generated required data from survey findings.

• In line 134 the author have said to use quartile rankings to divide wealth index and only divided in only 3 groups, moreover later in results have presented 5 categories in wealth index.

Result:

• The tables are difficult to understand as there may be mathematical issues just for instance the in table 1 out of 12783 participants 35.3 % are current tobacco user which comes to be around 4513, however while presenting the data of current smokeless tobacco user among current tobacco user the total number given is 5128 which comes to be around 40.1% of 12783 beyond the CI presented before.

Discussion:

• No findings or comparisons from other studies are represented showing lack of coherence.

Reviewer #2: The topic of the study is very interesting and the researchers seems to have put a lot of work into it. The overall idea is very good and explores an important issue in public health. However, I believe there are corrections that need to be made so that the manuscript comes out better. The suggestions are as follows:

1. General Comment:

-There are some inconsistencies in language in some places, please proof read the document again.

2. Introduction:

-It seems inappropriate to keep information from GATS 2014 of Pakistan and GATS 2017 of Bangladesh in the ‘Introduction section’. As you have included both of these surveys in your analysis and included sex and age as the factors while assessing disparities in SLT, this information is appropriate for the ‘Result section’.

3. Methodology:

- While giving operational definitions, there are repetitions in some places. Please try and remedy this.

4. Results:

-Adjusted Odd’s ratio(AOR) is mentioned only for some variables. In order to maintain consistency, it may be better to mention the AOR for all the variables mentioned in the writing of results section.

-Statistical tests that have been used are not very clear in the tables.

5. References:

-Some references do not seem to follow the journal guidelines. Please make needed correction.

6. PLOS authors have the option to publish the peer review history of their article (what does this mean?). If published, this will include your full peer review and any attached files.

Reviewer #1: **Yes: **Mukesh Poudel

Reviewer #2: No

---

## [Author Response · Author response to Decision Letter 0]

28 Feb 2021

EDITOR COMMENTS

Comment 1: Production, import and sale of illicit smokeless tobacco products are also contributory factors fueling the SLT epidemic in these countries. Although the objectives of this study does not directly relate to it, it would be worthwhile to mention this in the discussion.

Response to Comment 1: We have incorporated this in our discussion section. which reads as follows (See page 5): “The challenge to addressing tobacco use among low SES populations in Bangladesh, India, and Pakistan may also need to address the existence of tax evasion, illicit sales, and production of smokeless tobacco, including illicit trade and low levels of taxation for these products (11, 24). This illicit supply chain provides more affordable and accessible SLT products, which can sustain the consumption of these products among individuals with lower SES. Addressing these issues could help diminish the extent to which critical populations initiate and use these products, including those with low SES population and young people.” 

REVIEWER 1

Comment 1: In line 71 and 73 the disease burden due to SLT use is explained in DALYs lost. It is confusing as one DALY means a year of healthy life lost. For eg: disease burden due to SLT use was 8,691,827 disability-71 adjusted life years (DALYs) and 72 348,798 deaths sounds appropriate to me.

Response to Comment 1: We have revised this section and now reads as follows (See page X): “Sinha et al. (6) estimated that globally, the number of deaths that could be attributed to SLT, due to all causes, was 652,494 (234,008–1,081,437). This estimate excludes Europe, where studies showed no statistically significant association between all-cause mortality and SLT use. South East Asian Region carried the major proportion (88%) of this burden (6).”

Comment 2: In line 98 author have said no literature have examined disparities in SLT use however in a paragraph just above in line 93 to 97 have presented the findings from GATS survey which have found different disparities in SLT use.

Response to Comment 2: We revised to reads as follows (see page X): “However, there are limited studies that have examined disparities in the use of specific SLT products, including disparities by demographics and socioeconomic status. In addition, there are few studies on SLT use disparities associated with exposure to SLT-specific marketing, SLT-specific warnings, or self-reported awareness of SLT harms, which could influence use of these products.” 

Comment 3: The authors have stated using the data from GATS survey at different time period in three south Asian countries. In line 111 the sample size and response rate of the study was according to the GATS survey itself, or authors identified is confusing to understand. The data collection method is not clear, how the data were extracted for analysis and presentation needs a clearer explanation.

Response to Comment 3: The text has been revised as follows (See page 4): “This study utilized GATS data from Bangladesh, India, and Pakistan. In all three countries, GATS was implemented as a nationally representative, household-based, cross-sectional survey of non-institutionalized adults aged 15 years or older, which uses a standardized sample design, survey protocols, and questionnaire to ensure data comparability (17). GATS was conducted in Bangladesh in 2017 with 14,880 completed individual interviews for a 90.8 overall response rate; India in 2016-2017 with 74,037completed individual interviews for a 92.9% overall response rate, and Pakistan in 2014 with 7,831 completed individual interviews for an overall response rate of 81.0%. 

Comment 4: Different measures are explained as means of data collection but were these measures an operational definition from GATS survey or authors of the study created it? If it is from GATS survey than it is to be explained as the measures taken in survey to generate data in different variables were: - OR if it came from author of this article it should explain on more how data are presented in GATS survey and how authors generated required data from survey findings.

 Response to Comment 4: The manuscript has been revised as follows (see page 5):

Tobacco Product Use

Current SLT use was defined with the question, “Do you currently use smokeless tobacco on a daily basis, less than daily, or not at all. Smokeless tobacco include zarda, sada pata, gul, khoinee, nosshil (Bangladesh); tobacco leaf, betel quid with tobacco, sada/surti, khaini or tobacco lime mixture, gutkha, pan masala with zarda, mawa, gul, gudaku, mishri (India); and naswar, nass (sniffed in the nose), paan with tobacco, gutka, mainpuri and others (Pakistan)”. Using this question, we develop a new “current SLT use” variable consisting of individuals using SLT daily and less than daily. 

Current tobacco smoking was defined with the question, “Do you currently smoke tobacco on a daily basis, less than daily, or not at all?” Using the question, we develop a new “current tobacco smoking” variable consisting of individuals who smoke tobacco on a daily and less than daily basis. 

Current tobacco use was defined using the definition advanced by Kar, Sivanantham, Chinnakali, and Thiagarajan (18): “Do you currently smoke tobacco?” and “Do you currently use smokeless tobacco?” Those who responded as “daily” and “less than daily” to both or either one of the questions, were defined as “current tobacco user” and those who responded, “not at all” were defined as “current tobacco nonuser” (18).

Comment 5: In line 134 the author have said to use quartile rankings to divide wealth index and only divided in only 3 groups, moreover later in results have presented 5 categories in wealth index.

Response to Comment 5: We have revised the manuscript as follows (see page 6): “The wealth index was divided into wealth index quintile rankings as follows: lowest, low, middle, high, and highest (19).”

Comment 6: The tables are difficult to understand as there may be mathematical issues just for instance the in table 1 out of 12783 participants 35.3 % are current tobacco user which comes to be around 4513, however while presenting the data of current smokeless tobacco user among current tobacco user the total number given is 5128 which comes to be around 40.1% of 12783 beyond the CI presented before.

Response to Comment 6: We verified the numbers in the tables, and they are correct. We have also clarified the labeling in table 1. Specifically, the N (sample size) revised to “n” is unweighted and the Percent (95% CI) are weighted estimates. Thus, the number of respondents (unweighted n) multiplied by the weighted percent of current tobacco users would not be expected to align. This approach is commonly used in the literature, and thus, have retained this presentation of the results. 

However, to help further clarify this issue for the reviewer, we have provided the below table presenting unweighted and weighted estimates. 

Current smokeless tobacco users among tobacco users - GATS Bangladesh 2017

 Unweighted percent1 Unweighted number Weighted percent Weighted number2

Total (Tobacco users) 100 5,128 100 37,761,628

Current smokeless tobacco users 60.82 3,119 58.38 22,047,040

Non-current smokeless tobacco users 39.18 2,009 41.62 15,714,588

Comment 7: Discussion: No findings or comparisons from other studies are represented showing lack of coherence.

Response to Comment 7: We have added comparison and reference to other studies in the discussion and now reads as follows (see page 14): “Additionally, all three countries showed a marked socio-demographic and economic disparity in SLT use defined by sex, education, wealth index, and age. Specifically, adults with lower education levels, and adults with lower wealth index had significantly higher odds of current SLT use. These findings are consistent with those in other studies, where use of SLT was high among those with low SES(12, 22, 23).”

REVIEWER 2

Comment 1: General Comment: There are some inconsistencies in language in some places, please proof read the document again.

Response to Comment 1: We have reviewed, edited, and proofed the manuscript to remove inconsistences.

Comment 2: Introduction: It seems inappropriate to keep information from GATS 2014 of Pakistan and GATS 2017 of Bangladesh in the ‘Introduction section’. As you have included both of these surveys in your analysis and included sex and age as the factors while assessing disparities in SLT, this information is appropriate for the ‘Result section’.

Response to Comment 2: We present the previously published findings from GATS Pakistan (2014) and GATS Bangladesh (2017) to provide background, and to reinforce the gaps in the scientific literature filled by the present study. Since these estimates were conducted and published elsewhere in the scientific literature, we do not feel it is appropriate to cite these data in our own Results. Instead, we duly acknowledge these data in the Introduction, while using it as a foundation to reinforce the objective of the present study, which provides further detail on this data source and the presented measures. 

Comment 3: Methodology: While giving operational definitions, there are repetitions in some places. Please try and remedy this.

Response to Comment 3: We have revised the operational definitions to address the repetitions, which now reads as follows (See page 5):

Tobacco Product Use

Current SLT use was defined with the question, “Do you currently use smokeless tobacco on a daily basis, less than daily, or not at all. Smokeless tobacco include zarda, sada pata, gul, khoinee, nosshil (Bangladesh); tobacco leaf, betel quid with tobacco, sada/surti, khaini or tobacco lime mixture, gutkha, pan masala with zarda, mawa, gul, gudaku, mishri (India); and naswar, nass (sniffed in the nose), paan with tobacco, gutka, mainpuri and others (Pakistan)”. Using this question, we develop a new “current SLT use” variable consisting of individuals using SLT daily and less than daily. 

Current tobacco smoking was defined with the question, “Do you currently smoke tobacco on a daily basis, less than daily, or not at all?” Using the question, we have developed a new “current tobacco smoking” variable consisting of individuals who smoke tobacco on a daily and less than daily basis. 

Current tobacco use was defined using the definition advanced by Kar, Sivanantham, Chinnakali, and Thiagarajan (18): “Do you currently smoke tobacco?” and “Do you currently use smokeless tobacco?” Those who responded as “daily” and “less than daily” to both or either one of the questions, were defined as “current tobacco user” and those who responded, “not at all” were defined as “current tobacco nonuser” (18).

Comment 4: Results: Adjusted Odd’s ratio(AOR) is mentioned only for some variables. In order to maintain consistency, it may be better to mention the AOR for all the variables mentioned in the writing of results section.

Response to Comment 4: We have updated the results section to include aOR for all the variables mentioned in the writing of results section. The section reads as follows (see page 12-13):

Multivariate Analyses 

The odds of SLT use was higher among females than males in Bangladesh (aOR: 1.53; CI: 1.31-1.80), but lower in females than males in India (aOR: 0.26; CI: 0.24-0.28) and Pakistan (aOR: 0.20; CI: 0.14-0.28). Compared to those aged 15-24 years, the odds of SLT use in Bangladesh was higher among those aged 25-45 years (aOR: 4.30; CI: 3.34-5.62), 45-65 years (aOR: 9.80; CI: 7.40-12.86), and 65+ years (aOR: 12.80 (9.15-17.96). In India, compared to those aged 15-24 years, the odds of SLT use was higher among those aged 25-45 years (aOR: 2.40; CI: 2.19-2.73), 45-65 years (aOR: 2.70; CI: 2.33-3.04), and 65+ years (aOR: 2.8 (2.36-3.24). In Pakistan, compared to those aged 15-24 years, the odds of SLT use was higher among those aged 25-45 years (aOR: 2.40; CI: 1.47-3.76), 45-65 years (aOR: 3.70; CI: 2.23-6.20), and 65+ years (aOR: 2.20, CI: 1.15-4.19). 

In Bangladesh, compared to those with no formal education, the odds of SLT use were lower among those with less than primary aOR: 0.78; CI 0.65-0.93), primary completed (aOR: 0.59; CI: 0.48-0.73), less than secondary (aOR: 0.45; CI:0.36-0.56), secondary/high school complete (aOR:0.32; CI: 0.23-0.44), and college and above (aOR: 0.23; CI: 0.14-0.39). In India, compared to those with no formal education , the odds of SLT use were lower among those with primary completed (aOR: 0.84; CI: 0.75-0.93), less than secondary (aOR: 0.75; CI:0.67-0.83), secondary/high school complete (aOR:0.46; CI: 0.41-0.52), and those with college and above (aOR: 0.27; CI: 0.23-0.33). In Pakistan, compared to those with no formal education, the odds of SLT use were lower among those with secondary/high school complete (aOR:0.43; CI: 0.29-0.64), and those with college and above (aOR: 0.26; CI: 0.11-0.61). 

Those ranked in the highest wealth index in all three countries had lower odds of SLT use compared to those in the lowest wealth index. In Bangladesh, compared to the lowest wealth index, the odds of SLT use were lower among those in the low (aOR:0.71; CI: 0.59-0.85), medium (aOR: 0.75; CI:0.62-0.91), high (aOR: 0.63: CI: 0.50-0.78), and highest (aOR: 0.44; CI: 0.33-0.58) wealth indices. In India, compared to the lowest wealth index, the odds of SLT use were lower among those in the low (aOR:0.69; CI: 0.63-0.76), medium (aOR: 0.59; CI: 0.52-0.66), high (aOR: 0.42: CI: 0.37-0.48), and highest (aOR: 0.25; CI: 0.21-0.30) wealth indices. In Pakistan, compared to the lowest wealth index, the odds of SLT use were lower among those in the low (aOR: 0.63; CI: 0.42-0.93), medium (aOR: 0.56; CI: 0.34-0.91), and highest (aOR: 0.3; CI: 0.15-0.55) wealth indices. 

In all three countries, the odds of SLT use were lower among current tobacco smokers compared to non-tobacco smokers. Compared to non-tobacco smokers, the odds of SLT use among current tobacco smokers was lower in Bangladesh (aOR: 0.65; CI: 0.54-0.80), India (aOR: 0.73; CI: 0.65-0.82), and Pakistan (aOR: 0.39; (0.24-0.65). 

Comment 5: Statistical tests that have been used are not very clear in the tables.

Response to Comment 5: The statistical tests are specified in the footnotes for the tables. 

In Table 1, the statistical test footnotes are: * Wild & Seber tests p<0.05 compared to that of SLT use from the same country; and ** t-tests p<0.05 Compared to that of females from the same country. 

In Table 2, the statistical tests presented in the footnotes are: REF: Reference group. aOR: Adjusted odds ratio. CI: Confidence Interval. * p value for Wald F test p<0.05 from logistic regression. # Chi square tests p<0.05.$ Cochran-Mantel-Haenszel tests for trend p<0.05. SLT: Smokeless tobacco

Comment 6: References: Some references do not seem to follow the journal guidelines. Please make needed correction.

Response to Comment 6: We have revised the references according to Plos One guidelines.

---

## [Decision Letter · Decision Letter 1]

1 Apr 2021

Disparities in Smokeless Tobacco Use in Bangladesh, India, and Pakistan: Findings from the Global Adult Tobacco Survey, 2014 -2017

PONE-D-21-01174R1

Dear Dr. Mbulo,

We’re pleased to inform you that your manuscript has been judged scientifically suitable for publication and will be formally accepted for publication once it meets all outstanding technical requirements.

Kind regards,

Pranil Man Singh Pradhan

Academic Editor

PLOS ONE

Additional Editor Comments (optional):

Reviewers' comments:

Reviewer's Responses to Questions

**Comments to the Author**

1. If the authors have adequately addressed your comments raised in a previous round of review and you feel that this manuscript is now acceptable for publication, you may indicate that here to bypass the “Comments to the Author” section, enter your conflict of interest statement in the “Confidential to Editor” section, and submit your "Accept" recommendation.

Reviewer #1: All comments have been addressed

Reviewer #2: All comments have been addressed

2. Is the manuscript technically sound, and do the data support the conclusions?

Reviewer #1: Yes

Reviewer #2: Yes

3. Has the statistical analysis been performed appropriately and rigorously? 

Reviewer #1: Yes

Reviewer #2: Yes

4. Have the authors made all data underlying the findings in their manuscript fully available?

Reviewer #1: (No Response)

Reviewer #2: Yes

5. Is the manuscript presented in an intelligible fashion and written in standard English?

Reviewer #1: (No Response)

Reviewer #2: Yes

6. Review Comments to the Author

Reviewer #1: Dear Authors.

The paper was interesting to go through. I somewhere found different grammatical issues, so may need a review again before publishing. The abbrevations used are somewhere SLT and somewhere STL, is it same or different.

Reviewer #2: I agree with your replies to most of my previous comments. However, I am still unconvinced about your reply to comment 2 and comment 5

Regarding comment 2: I still stand by my previous comment.

Regarding comment 5: In Table 1, I can appreciate that the tests have been indicated. However, as there are many variables in the table. It is confusing which variables are being assessed by the statistical tests.

7. PLOS authors have the option to publish the peer review history of their article (what does this mean?). If published, this will include your full peer review and any attached files.

Reviewer #1: **Yes: **Mukesh Poudel

Reviewer #2: No

---

## [Editor Report · Acceptance letter]

12 Apr 2021

PONE-D-21-01174R1 

Disparities in Smokeless Tobacco Use in Bangladesh, India, and Pakistan: Findings from the Global Adult Tobacco Survey, 2014 -2017 

Dear Dr. Mbulo:

I'm pleased to inform you that your manuscript has been deemed suitable for publication in PLOS ONE. Congratulations! Your manuscript is now with our production department. 

Kind regards, 

on behalf of

Dr. Pranil Man Singh Pradhan 

Academic Editor

PLOS ONE